# Enhancement of Triple-Negative Breast Cancer-Specific Induction of Cell Death by Silver Nanoparticles by Combined Treatment with Proteotoxic Stress Response Inhibitors

**DOI:** 10.3390/nano14191564

**Published:** 2024-09-27

**Authors:** Christina M. Snyder, Beatriz Mateo, Khushbu Patel, Cale D. Fahrenholtz, Monica M. Rohde, Richard Carpenter, Ravi N. Singh

**Affiliations:** 1Department of Cancer Biology, Wake Forest University School of Medicine, Winston-Salem, NC 27157, USA; christinasnyder0@gmail.com (C.M.S.); bmateo@wakehealth.edu (B.M.); khpatel@wakehealth.edu (K.P.); cfahren@highpoint.edu (C.D.F.); monicarohde5@gmail.com (M.M.R.); 2Fred Wilson School of Pharmacy, High Point University, High Point, NC 27268, USA; 3Department of Biochemistry and Molecular Biology, Indiana University School of Medicine, Bloomington, IN 47405, USA; richcarp@iu.edu; 4Atrium Health Wake Forest Baptist Comprehensive Cancer Center, Winston-Salem, NC 27157, USA

**Keywords:** nanotoxicity, heat shock, proteotoxicity, integrated stress response, cancer therapy

## Abstract

Metal nanoparticles have been tested for therapeutic and imaging applications in pre-clinical models of cancer, but fears of toxicity have limited their translation. An emerging concept in nanomedicine is to exploit the inherent drug-like properties of unmodified nanomaterials for cancer therapy. To be useful clinically, there must be a window between the toxicity of the nanomaterial to cancer and toxicity to normal cells. This necessitates identification of specific vulnerabilities in cancers that can be targeted using nanomaterials without inducing off-target toxicity. Previous studies point to proteotoxic stress as a driver of silver nanoparticle (AgNPs) toxicity. Two key cell stress responses involved in mitigating proteotoxicity are the heat shock response (HSR) and the integrated stress response (ISR). Here, we examine the role that these stress responses play in AgNP-induced cytotoxicity in triple-negative breast cancer (TNBC) and immortalized mammary epithelial cells. Furthermore, we investigate HSR and ISR inhibitors as potential drug partners to increase the anti-cancer efficacy of AgNPs without increasing off-target toxicity. We showed that AgNPs did not strongly induce the HSR at a transcriptional level, but instead decreased expression of heat shock proteins (HSPs) at the protein level, possibly due to degradation in AgNP-treated TNBC cells. We further showed that the HSR inhibitor, KRIBB11, synergized with AgNPs in TNBC cells, but also increased off-target toxicity in immortalized mammary epithelial cells. In contrast, we found that salubrinal, a drug that can sustain pro-death ISR signaling, enhanced AgNP-induced cell death in TNBC cells without increasing toxicity in immortalized mammary epithelial cells. Subsequent co-culture studies demonstrated that AgNPs in combination with salubrinal selectively eliminated TNBCs without affecting immortalized mammary epithelial cells grown in the same well. Our findings provide additional support for proteotoxic stress as a mechanism by which AgNPs selectively kill TNBCs and will help guide future efforts to identify drug partners that would be beneficial for use with AgNPs for cancer therapy.

## 1. Introduction

Proteotoxicity is a condition caused by the accumulation of misfolded and aggregated proteins in the cell and can lead to cell death if stress conditions are prolonged or unmitigated [1]. To alleviate proteotoxic stress, misfolded proteins are sequestered and delivered to proteasomes, lysosomes, and autophagosomes to be degraded [2]. Some cancer types exhibit high baseline levels of proteotoxic stress compared to non-cancer cells, and this may be an exploitable weakness [3]. For example, studies by Gupta et al. revealed that a subset of triple-negative breast cancer cells (TNBC) are vulnerable to small-molecule drugs, including tunicamycin and thapsigargin, which increased the accumulation of misfolded proteins [4,5]. These TNBCs are sensitive because they synthesized and secreted large amounts of extracellular matrix proteins, generating an enormous burden on the protein quality-control machinery. However, tunicamycin and thapsigargin have thus far failed to be effective clinically due to dose-limiting toxicities [6,7].

Nanotechnology offers the possibility to increase the efficacy and reduce off-target toxicity of small-molecule drugs used as proteotoxic stress inducers [8], but difficulty with formulation, characterization, and scale-up have hampered most efforts to translate drug-carrying nanoparticles for cancer therapy in humans [9]. The core technologies of clinically approved nanotherapeutics remain largely unchanged since the development of Doxil^®^ [10]. Nanoparticles made of metals, including gold, silver, iron, and gadolinium, have been tested for therapeutic and imaging applications in pre-clinical models of cancer [11], but fears of toxicity have limited their translation. An emerging concept in nanomedicine is exploiting the inherent drug-like and cytotoxic properties of unmodified, metal nanomaterials for cancer therapy [12,13,14,15,16]. To be useful clinically, there must be a window between the toxicity of the nanomaterial to cancer and toxicity to normal cells. This necessitates identification of specific vulnerabilities in cancers that can be targeted using nanomaterials without inducing off-target toxicity. For example, TNBC cells, but not immortalized mammary epithelial cells, are sensitive to silver-nanoparticle (AgNP)-induced proteotoxic stress, as indicated by accumulation of protein aggregates, increased protein oxidation, and activation of proteotoxic stress-signaling responses [17,18,19,20,21,22]. Use of AgNPs (or other nanometals) themselves as the therapeutic agent simplifies formulation, characterization, and scale-up compared to conventional drug-loaded nanoparticles [23]. We hypothesize that the therapeutic efficacy of AgNPs could potentially be enhanced by co-delivery of drugs that modulate the proteotoxic stress response, but care must be taken to examine toxicities of therapeutics that will be administered together to ensure that off-target toxicity profiles are not increased.

The heat shock response (HSR) plays a key role in mitigating proteotoxicity. Heat shock proteins (HSPs) are a family of cytosolic, nuclear, and membrane-bound proteins that function as chaperones by assisting in protein folding, localization, and transport [24]. Activity of HSPs enables cells to endure proteotoxicity induced by stresses, such as heat, oxidative stress, nutrient deprivation, and impaired protein degradation machinery. HSPs include stress-inducible heat shock protein 70 (HSP70) and heat shock protein 27 (HSP27), as well as the ubiquitously expressed heat shock protein 90 (HSP90) [25]. Under normal conditions, heat shock factor 1 (HSF1), known as the ‘master heat shock regulator’, is present as an inactive monomer bound to HSP90 or HSP70 in the cytosol of cells [25]. When misfolded proteins are abundant, HSF1 dissociates from HSP70 or HSP90, trimerizes, autophosphorylates, and translocates to the nucleus [26]. HSF1 then binds heat shock elements (HSEs) on DNA to induce transcription of HSPs, which bind unfolded or misfolded proteins and act as chaperones to repair damaged proteins, when possible, or sequester damaged proteins to enable subsequent degradation [25]. When not bound to misfolded proteins, HSPs regulate HSF1 activity and can inhibit HSF1 activation or induce HSF1 degradation through multiple feedback inhibitory loops [27,28,29]. HSPs are frequently overexpressed in cancer, may promote malignant phenotypes, and HSR inhibitors have been investigated as cancer therapeutics [30]. Inhibition of the HSR increases sensitivity to proteotoxic stress, but the combination of AgNPs and HSR inhibitors has not been tested.

A second mechanism cells use to mitigate proteotoxicity is the integrated stress response (ISR). The ISR is initiated by stress-sensing kinases: double-stranded RNA-dependent protein kinase (PKR), PKR-like ER kinase (PERK), heme-regulated eIF2α kinase (HRI), and general control non-de-repressible 2 (GCN2) [31]. Each is activated by different types of stress, but all similarly initiate the ISR through phosphorylation of the alpha subunit of eukaryotic translation initiation factor 2 (eIF2α) on serine 51. Phosphorylation of eIF2α causes a decrease in global mRNA translation to reduce the protein synthesis burden, and simultaneously activates an alternative transcriptional program needed to boost protein folding and degradative capacity. Paradoxically, prolonged phosphorylation of eIF2α increases expression of C/EBP Homologous Protein (CHOP), which drives pro-apoptotic pathways. We and others observed phosphorylation of eIF2α in response to AgNPs in TNBC cells, but not in normal breast cells [17,20,22]. Because sustained ISR signaling initiates cell death programs [2,31], cells activate a signaling feedback that increases expression of Growth Arrest and DNA-damaged protein 34 (GADD34) and the serine/threonine protein phosphatase 1, which form a complex (GADD34:PP1) that dephosphorylates eIF2α to restore normal transcriptional programs [31]. Salubrinal is a drug that inhibits the phosphatase activity of the GADD34:PP1 complex, thereby sustaining ISR signaling. Because the ISR can produce both protective and lethal effects, use of salubrinal on cells under proteotoxic stress has the potential to be either protective or to increase cell death [32,33,34,35]. It is not known if salubrinal will protect against or enhance AgNP-induced cell death.

Here, we examine two types of proteotoxic stress responses, the HSR and the ISR, following AgNP exposure in TNBC and immortalized mammary epithelial cells. Furthermore, we employ drugs that modulate the HSR or ISR to identify potential pharmacologic partners that could increase the toxicity of AgNPs in TNBC without increasing toxicity in normal cells.

## 2. Materials and Methods

*Silver nanoparticles:* Here, 25 nm spherical AgNPs stabilized with polyvinylpyrrolidone (PVP) were purchased as dried powders from nanoComposix, Inc. (San Diego, CA, USA). The manufacturer specified a mass ratio of 17% Ag to 83% PVP for the nanoparticles. Nanoparticles were dispersed by bath sonication in phosphate-buffered saline (PBS), pH 7.4, without calcium or magnesium (Invitrogen, Carlsbad, CA, USA), at a concentration of 5 mg/mL based upon the mass of silver contained in the nanoparticles (i.e., excluding PVP), and were then diluted in cell culture medium to the final concentration listed in the figures prior to addition to wells containing cells. The physicochemical properties of this material were characterized previously [36]. Briefly, characteristics of the prepared stock of AgNPs were as follows: hydrodynamic diameter in PBS at pH 7.4: 36.5  ±  0.7 nm; ζ-potential in PBS at pH 7.4: −15.5  ±  1.6 mV; plasmon resonance peak in PBS at pH 7.4: 402 nm; soluble silver (Ag^+^) present in the AgNP suspension: less than 0.001% Ag^+^ by mass.

*Cell culture:* BT549, MDA-MB-231, and MCF-10A cells were purchased from the ATCC (Manassas, VA, USA). SUM159 cells were purchased from Asterand (now BioIVT, Westbury, NY, USA). iMEC cells were provided by Dr. Elizabeth Alli in the Department of Cancer Biology at Wake Forest University School of Medicine. Cell lines were expanded, and low-passage stocks were stored in liquid nitrogen and maintained by the Wake Forest Comprehensive Cancer Center Cell Engineering Shared Resource. Growth media for these cell lines are listed in Table 1. Cells were verified to be free from mycoplasma contamination by routine testing using the MycoAlert Mycoplasma Detection Kit (Lonza, Basel, Switzerland). Cells were passaged, and the medium was changed twice weekly. Cell monolayers were grown on tissue-culture-treated plastics purchased from Corning Life Sciences (Corning, NY, USA). Cells were maintained in culture for no longer than 4 months before new cultures were established from low-passage frozen stocks.

*MTT assay*: Cells were seeded on 96-well plates at a density of 5000 cells per well in 100 μL of complete media, recovered for 24 h, and then were exposed to combinations of AgNPs, KRIBB11 (Selleck Chemicals, Houston TX, USA), or salubrinal (MedChemExpress, Monmouth Junction, NJ, USA) in 100 μL of complete media containing doses of each drug, as listed in the figures. After 72 h, media containing treatments were replaced with 200 μL of media containing 0.5 mg/mL of 3-(4,5- dimethylthiazol-2-yl)-2,5-diphenyltetrazolium bromide (MTT; Sigma-Aldrich, St. Louis, MO, USA) and incubated for 1 h at 37 °C. After that, media were removed, cells were lysed in 200 μL of dimethyl sulfoxide (Thermo Fisher Scientific, Pittsburgh, PA, USA), and absorbance was read using a Molecular Devices Emax Precision Microplate Reader (San Jose, CA, USA) at 560 nm. Absorbance measurements at 650 nm were subtracted from each reading to correct for any inconsistencies in optical properties between wells.

*Western Blotting:* Cells were plated on 6 cm dishes at a density of 1,000,000 cells in 4 mL of complete media. Cells were allowed to recover for 48 h and then were exposed to AgNPs with or without salubrinal for 24 h at 37 °C. Medium was removed and cells were washed twice with ice-cold phosphate-buffered saline before lysis using Mammalian Protein Extraction Reagent supplemented with Halt Protease and Phosphatase Inhibitor Cocktail (both from Thermo Fisher Scientific). The protein concentration was determined for each sample using a Pierce bicinchoninic acid (BCA) protein assay kit (Thermo Fisher Scientific), according to the manufacturer’s instructions. Proteins were size fractionated by gel electrophoresis and then transferred to a nitrocellulose membrane (Thermo Fisher Scientific). Non-specific binding was blocked by incubation for 30 min at room temperature with tris-buffered saline containing 0.1% Tween-20 (TBS-T; Bio-Rad) and either 5% blotting-grade blocker (Bio-Rad) or 5% bovine serum albumin (BSA; Sigma-Aldrich). Membranes were incubated overnight at 4 °C in dilutions containing TBS-T and either 5% blotting-grade blocker or 5% BSA and primary antibody. Primary antibodies used included the following: phospho(Ser51)-eIF2α (9721), eIF2α (5324), CHOP (2895), caspase-3 (9662), caspase-7 (9492), caspase-9 (9502), HSF1 (4356), HSP90 (4874), HSP70 (4876), phospho(Ser78)-HSP27 (2405), HSP27 (2402), GAPDH (2118), and β-actin (4970), purchased from Cell Signaling Technologies (Danvers, MA, USA), and phospho(Ser326)-HSF1 (ab76076) purchased from Abcam (Waltham, MA, USA). Membranes were washed and then incubated for 1 h at room temperature with anti-rabbit (Cell Signaling Technologies) or anti-mouse (Cell Signaling Technologies) horseradish peroxidase (HRP)-conjugated secondary antibodies diluted in 5% blotting-grade blocker in TBS-T. Immunoreactive products were visualized by chemiluminescence using SuperSignal Pico West reagent (Thermo Fisher Scientific).

*Heat shock reporter assay*: Cells were seeded on 24-well plates at a density of 100,000 cells per well in 1 mL media and allowed to recover for 2 days. Medium was replaced with 1 mL of fresh medium without penicillin or streptomycin. pGL4.41(luc/HSE/hygro) (Promega, Madison, WI, USA) was mixed with optiMEM (Thermo Fisher Scientific) and Xtreme Gene 9 (Sigma Aldrich) at a ratio of 33:3:1 (media (vol.):lipid (vol.):DNA (mass)). After 15 min, 0.2 μg DNA complexed to Xtreme Gene 9 was added to each well. After 24 h, cells were treated with AgNPs, then exposed to heat shock (43 °C) or physiologic temperature (37 °C) by incubation for 1 h in medium warmed to the respective temperature. After recovery for 3 h at 37 °C, luciferase activity was assessed using a commercial assay kit (Promega Luciferase Assay System), as per the manufacturer’s instructions. Briefly, cells were harvested in reporter lysis buffer and centrifuged for 5 min at 4 °C at 15,000 RPM in a microcentrifuge. Lysates were transferred to a white 96-well plate (Greiner Bio-One, Monroe, NC, USA), luciferin was added, and luminescence was evaluated using a FLUOstar Optima plate reader (BMG Labtech, Cary, NC, USA). Total protein in lysates was estimated using the Pierce BCA protein assay kit (Thermo Fisher Scientific), as per the manufacturer’s instructions. Luminescence in each well was normalized based on the protein concentration.

*Co-culture and flow cytometry:* BT549 cells expressing mCherry (BT549^mCherry^) and iMEC cells expressing GFP (iMEC^GFP^) were produced by lentiviral transduction using concentrated lentivirus vectors produced by the Cell Engineering Shared Resource at Atrium Health Wake Forest Baptist Comprehensive Cancer Center. Briefly, pLenti-CMV-GFP-Puro-LV (a gift from Eric Campeau and Paul Kaufman; Addgene plasmid #17448; http://n2t.net/addgene:17448; RRID:Addgene_17448) [37] or pCDH-CMV-mCherry-T2A-Puro-LV (a gift from Kazuhiro Oka; Addgene plasmid #72264; http://n2t.net/addgene:72264; RRID:Addgene_72264) were co-transfected into HEK293T cells (ATCC) with plasmids from the pPACKH1 HIV Lentivector Packaging Kit (Systems Biosciences, Palo Alto, CA, USA), according to the manufacturer’s instructions. Viral supernatants were concentrated using the Lenti-X Concentrator Kit (Takara Biosciences USA, San Jose, CA, USA) according to the manufacturer’s instructions. Infectious lentivirus titers were determined in HEK293T cells, and cell lines were transduced at a multiplicity of infection of 10 and selected for 2 weeks in puromycin. For co-culture studies, cells were plated in 6-well plates at a density of 100,000 BT549^mCherry^ and 100,000 iMEC^GFP^ cells per well in 4 mL of complete iMEC media and allowed to adhere overnight. The next day, cells were treated with a range of AgNP doses with or without salubrinal. Medium was changed 3 days after treatments, and cells were grown in the absence of AgNPs or salubrinal drugs for 5 days. Cells were trypsinized, washed, diluted to a concentration of 10^6^ cells per mL in PBS, and fixed in paraformaldehyde (1% final concentration). The percentage of mCherry (excitation: 561, emission: 610/20) and GFP (excitation: 488, emission: 530/30) expressing cells was quantified by flow cytometry using an LSRFortessa X-20 (BD Biosciences, San Jose, CA, USA), and data were analyzed using BD FACSDiva V6.0 software.

*Statistical Analysis:* Analysis was performed as described in the figure legends using Prism 9.1.3 software (GraphPad, San Diego, CA, USA). The number of technical and biological replicates for each experiment is included in the figure legends. Synergy was examined using the Chou–Talalay method [38]. Isobolograms at the 50% fraction affected dose (Fa = 0.5) were plotted using Excel.

## 3. Results

*AgNPs do not strongly induce a protective HSR and decrease HSPs in TNBC cells to a greater degree than in mammary epithelial cells*.

We verified our previous findings [17,22,36,39] of differences in the response of TNBCs (BT549, MDA-MB-231, and SUM159) and immortalized mammary epithelial cells (iMEC and MCF-10A) to AgNP-induced cytotoxicity. As shown in Figure 1A, the cytotoxic IC_50_ values of AgNPs were 5–10-fold less for TNBCs compared to immortalized mammary epithelial cells. We next investigated the effect AgNPs had on the heat shock response. Cells were treated with doses of AgNPs approximating the mean IC_50_ (18.75 µg/mL), twice the IC_50_ (37.5 µg/mL), and four times the IC_50_ (75 µg/mL) for AgNPs in TNBC cells. We noted dose-dependent decreases in phosphorylated (Serine 326) HSF1 and total HSP70 in both BT549 and MDA-MB-231 cells after AgNP exposure (Figure 1B). AgNP treatment also increased HSP27 phosphorylation in BT549 and MDA-MB-231 cells but decreased total HSP27 in BT549 cells. No change in pHSF1 was observed in AgNP-treated iMEC or MCF10A cells, but HSP70 decreased at high doses. No changes in pHSP27 or total HSP27 were seen in AgNP-treated iMEC or MCF10A cells. Neither HSP90 nor total HSF1 expression changed with AgNP treatment in any cell line.

These studies indicated that AgNPs decreased the levels of multiple proteins involved in the HSR in TNBCs. In contrast, AgNPs had only modest effects on HSP expression in immortalized mammary epithelial cells. To determine if the effects of AgNPs on the heat shock response in TNBCs were caused at the transcriptional level, we transduced BT549 cells with a plasmid vector containing four copies of a heat shock response element (HSE), to which pHSF1 trimers bind and drive transcription of a luciferase reporter gene, indicative of a heat shock response (Figure 2A). Then, 24 h after transduction, cells were treated with AgNPs overnight, exposed to hyperthermic (43 °C) or physiologic temperature (37 °C) conditions, and luciferase activity was quantified 3 h later, as shown schematically in Figure 2B. The results of this experiment are shown in Figure 2C. AgNPs alone did not significantly increase HSE-driven luciferase activity at any of the doses tested. Exposure of cells to hyperthermia significantly increased luciferase activity, and this was reduced by co-treatment with KRIBB11, a selective inhibitor of HSF1, indicating that increased luciferase activity was dependent upon HSF1. Luciferase activity in AgNP-treated BT549 cells that were subsequently exposed to hyperthermia was similar to that of cells that were treated with hyperthermia alone. These data showed that AgNPs did not strongly induce a heat shock response in BT549 cells but also did not inhibit the capacity of cells to induce a heat shock response at a transcriptional level. Overall, our results suggest that AgNPs did not induce a protective HSR in TNBC cells. Decreased expression of HSPs in AgNP-treated cells is not due to transcriptional repression of the HSR and may be due to increased turnover or degradation of HSPs at the protein level.


*Heat shock inhibitors synergistically increase the cytotoxicity of AgNPs in both TNBC and immortalized mammary epithelial cells.*


Next, we asked if the combination of AgNPs and KRIBB11 resulted in a synergistic interaction in treatment efficacy compared to the individual agents. To perform this analysis, cells must be treated with fractions or multiples of a constant dose ratio of the two agents fixed at a 1:1 ratio of IC_50_ for the individual agents, as required by the Chou–Talalay method for drug combination analysis [38]. The IC_50_ for AgNPs for these two cell lines was determined in Figure 1A, and we performed triplicate experiments using similar approaches to determine the IC_50_ of KRIBB11 in BT549 and iMEC cells (Figure 3A). Based upon these experiments, the concentration ratio needed to achieve a 1:1 ratio of the IC_50_ of AgNP (μg/mL):KRIBB11 (μM) was defined for each cell line as 3.5:1 for BT549 cells or 4.15:1 for iMEC cells. We treated cells with the individual agents or the fixed ratio combination of the two agents and plotted the survival curves for each treatment for BT549 (Figure 3B) and iMEC cells (Figure 3C). To the right of the survival curves, we plotted the combination dose of AgNPs and KRIBB11 that achieved a fraction affected (Fa) of 0.5 (indicating 50% loss of cell viability), together with the predicted additive isobole for AgNPs combined with KRIBB11. The data indicated that the combination of AgNPs and KRIBB11 was synergistic because the point representing Fa = 0.5 for the concentrations of concurrently administered AgNP and was below the additive isobole line connecting the IC_50_ of AgNPs and KRIBB11 used as single agents. We further calculated the combination index (CI) relative to Fa = 0.5. CI < 1 denotes a synergistic interaction, CI > 1 represents antagonism, and CI = 1 represents an additive interaction. The CIs for AgNPs combined with KRIBB11 for BT549 cells and iMEC cells were 0.38 and 0.32, respectively, indicating synergistic dose enhancement in both cases. This indicated that inhibition of the HSR can potentially increase both on- and off-target toxicity of AgNPs.


*Salubrinal potentiates AgNP toxicity in mesenchymal TNBC without affecting immortalized mammary epithelial cells.*


We wanted to know if inhibition of eIF2α dephosphorylation using salubrinal would protect against or enhance AgNP-induced cell death. Initially, we performed studies to determine the cytotoxic IC_50_ of salubrinal alone in TNBC cells (BT549 and SUM159) or immortalized mammary epithelial cells (iMEC and MCF10A). Due to the limited solubility of salubrinal in aqueous media, we were not able to treat cells at a high enough dose to obtain an accurate cytotoxic IC_50_. Across the dose range tested, we observed no significant differences in the cytotoxicity of salubrinal among the cell lines (Figure 4A). Because we were unable to determine the cytotoxic IC_50_ of salubrinal, we could not perform synergy analysis for the combination of salubrinal and AgNPs. Instead, we combined increasing doses of AgNPs with fixed doses of salubrinal (5 or 10 µM) to determine if this combination affected AgNP cytotoxicity. We found that fixed doses of salubrinal significantly increased the cytotoxicity of AgNPs in BT549 and SUM159 cells but did not substantially increase toxicity of AgNPs in iMEC or MCF10A cells (Figure 4B).

We investigated ISR and apoptosis signaling in BT549 cells treated with combinations of AgNPs and salubrinal for 24 h. We observed increased phosphorylation of eIF2α in BT549 cells exposed to the combination of AgNPs and salubrinal compared to the individual treatments (Figure 5). This is consistent with previous studies showing that salubrinal prevents dephosphorylation of eIF2α, resulting in increased peIF2α when cells are exposed to proteotoxic stressors [40]. Additionally, we found that the combination of salubrinal and AgNPs induced increased expression of downstream effectors of peIF2α-induced cell death, including pro-apoptosis protein CHOP and decreased expression of full-length caspase-3, caspase-7, and caspase-9, which is indicative of caspase cleavage.

These results implied that the combination of AgNPs and salubrinal could be effective for eliminating TNBCs without damaging normal breast cells. To assess this directly, we performed a co-culture experiment in which BT549 and iMEC cells were grown together in a single well and treated with AgNPs, salubrinal, or a combination of the two. To distinguish between cell types, BT549 cells were transduced using a lentivirus to express mCherry, and iMEC cells were transduced to express eGFP. Cells were grown in co-culture at a 1:1 ratio, as shown in Figure 6A. The cells were treated with AgNPs, salubrinal, or a combination of AgNPs and salubrinal. The proportion of iMEC^GFP^ cells and BT549^mCherry^ cells that survived treatment was determined by flow cytometry, as shown in Figure 6B. In agreement with data obtained from monoculture studies, AgNPs alone decreased the percentage of BT549^mCherry^ cells in the co-culture, resulting in a concomitant increase in the percent of iMEC^GFP^ cells present. Salubrinal on its own had little effect, but the combination of AgNPs and salubrinal almost completely eliminated BT549^mCherry^ from the co-culture, while preserving iMEC^GFP^ cells. Quantification of results from triplicate experiments indicated that the surviving population was significantly depleted of BT549^mCherry^ cells (resulting in an increased fraction of iMEC^GFP^ cells) following AgNP exposure, and this effect was further enhanced when AgNPs were combined with salubrinal, as shown in Figure 6C. A portion (approximately 15%) of each cell line had low-level expression or did not express either fluorophore (seen in the lower left quadrant of each panel in the graphs in Figure 6B) and, therefore, the total percentage of cells quantified did not add to 100%. The results indicate that AgNPs alone selectively killed TNBC cells without affecting the growth of immortalized mammary epithelial cells grown together in co-culture, and this selectivity was enhanced when AgNPs were combined with salubrinal.

## 4. Discussion

AgNPs have enormous potential for clinical applications [11,17,20,22,36,39], but fears of toxicity have limited translation [41,42,43,44,45,46,47]. Our previous studies showed that AgNPs induced proteotoxicity and cell death in TNBC cells at doses that did not affect the homeostasis of normal breast epithelial cells in vitro [17,36] or in vivo [22]. Two key cell stress responses involved in mitigating proteotoxicity are the HSR and the ISR, but few studies have examined the roles of the HSR and ISR in mitigating or enhancing toxicities due to AgNP exposure. Here, we showed that the use of drugs that modulate these pathways can enhance the proteotoxic effects of AgNPs, and we identified combinations that are lethal to TNBC cells without increasing off-target toxicity on normal mammary epithelial cells.

Phosphorylation of HSF1 is indicative of HSR activation. Surprisingly, we observed decreased levels of pHSF1 in TNBC cells exposed to AgNPs, but no change in pHSF1 in immortalized mammary epithelial cells. We investigated if AgNPs affected the ability of HSF1 to transcriptionally activate the HSR. AgNPs alone did not significantly induce HSF1-dependent transcription, and AgNP-treated cells were capable of generating a robust HSF1-dependent transcriptional response after exposure to hyperthermia (43 °C). This indicated that AgNPs did not inhibit heat-induced transcriptional activity driven by pHSF1. Thus, our results suggested that decreased pHSF1 protein following AgNP exposure was due to increased turnover or degradation, possibly by a negative feedback inhibition loop [25,29]. We also observed decreased expression of HSP70 in both TNBC and immortalized mammary epithelial cells treated with AgNPs. This is in contrast to literature findings in other cell lines, indicating increased HSP70 following AgNP exposure [41,44,48,49,50]. It is unclear what factors drive these differences, but we have shown that Ag^+^ contamination in AgNP solutions can change physiological responses [36]. As discussed below, studies that fail to characterize Ag^+^ content or use AgNP solutions that contain Ag^+^ ions may be unreliable. HSP70 plays a key role in sequestering misfolded proteins, aids in delivering them to proteasomes, autophagosomes, and lysosomes, and may be degraded along with damaged proteins as cells attempt to recover from proteotoxic stress [51]. We also observed decreased total HSP27 in TNBCs treated with AgNPs. Similar to HSP70, HSP27 can increase the degradation of damaged proteins in stressed cells through the proteosomal pathway and is degraded in the process [52]. In contrast to total HSP27, we observed increased pHSP27 in AgNP-treated TNBC cells. Non-phosphorylated HSP27 forms long oligomers that mediate most of its chaperone functions. HSP27 phosphorylation leads to a significant decrease of the oligomeric size and reduces chaperone action. The smaller size of pHSP27 enables nuclear translocation of the cytosolic pool of HSP27, and this provides a mechanism for HSP27 to leave the cytosol and enter the nucleus, where it can be dephosphorylated and form oligomers with full chaperone capacity [53]. Therefore, a plausible hypothesis accounting for why AgNPs decrease total HSP27 and concomitantly increase pHSP27 expression is that pHSP27 increases in stressed cells but does not participate in chaperone functions and, therefore, does not enter protein degradation pathways.

In an effort to discover drug partners for AgNPs to enhance on-target and lower potential off-target toxic effects, we examined combinations of AgNPs with drugs that interfere with proteotoxic stress responses. We found that the HSF1 inhibitor, KRIBB11, synergistically increased the cytotoxicity of AgNPs in both TNBCs and immortalized mammary epithelial cells. These data indicated that the HSPs play a key role in mitigating AgNP toxicity, and inhibition of the HSR may increase off-target toxicity of AgNPs used for cancer therapy. We next tested salubrinal, an inhibitor of the eIF2α phosphatase complex [54], for its interaction with AgNPs. Inhibition of dephosphorylation of eIF2α using salubrinal enhanced pro-death ISR signaling and increased toxicity following AgNP treatment in TNBCs, but it had no effect on AgNP toxicity in immortalized mammary epithelial cells. When TNBC and immortalized mammary epithelial cells were treated in co-culture with a combination of salubrinal and AgNPs, the combination was substantially more toxic to TNBC cells than either AgNPs or salubrinal alone, but it had little effect on immortalized mammary epithelial cells. This is consistent with previous studies demonstrating that AgNP exposure increases peIF2α in TNBC cells but not in immortalized mammary epithelial cells [17,20,22,36]. Because ISR signaling is not activated by AgNPs in immortalized mammary epithelial cells, salubrinal cannot prolong ISR activation and drive cell death. Salubrinal has not been tested in humans, and further research will be needed to determine its translational potential. Despite this limitation, our results provide support to investigate the use of a combination of AgNPs and drugs that can sustain ISR signaling for treatment of TNBC.

Four distinct stress-sensing kinases (PKR, HRI, GCN2, and PERK) can initiate the ISR through phosphorylation of eIF2α [31]. In our study, we did not specifically determine which of these kinases drives the ISR in response to AgNPs. Previous studies linked AgNP exposure to activation of endoplasmic reticulum (ER) stress, as indicated by the unfolded protein response (UPR). Briefly summarized, the UPR is initiated by three ER-bound stress-sensing proteins, PERK, activating transcription factor 6α/β (ATF6), and inositol requiring enzyme 1α/β (IRE1). In unstressed states, the protein chaperone GRP78 binds and inhibits PERK, IRE1, and ATF6 on the ER membrane [55]. If the load of unfolded proteins is greater than the capacity of ER protein chaperones to sequester and fold, GRP78 dissociates from PERK, IRE1, and ATF6 to increase the folding capacity. This enables PERK and IRE1 to form homodimers and trans-autophosphorylate to initiate downstream signaling cascades, and frees ATF6 to translocate to the Golgi, where it is cleaved to its active form [56,57,58]. Although some studies showed that all three arms of the UPR are activated in retinal pigment cells [59] or in Chang liver and human lung fibroblasts [60] following AgNP exposure, other studies, including our own previous work [17,22,36], did not demonstrate activation of all arms of the UPR. In one study, AgNPs increased CHOP expression in drug-resistant breast cancer cells, but PERK dependence was not examined, nor was activation of the other arms of the UPR [19]. In another study, AgNP exposure increased CHOP and total IREα, but no evidence of phosphorylation of PERK, phosphorylation of IREα, cleavage of ATF6, or activation of other downstream signaling cascades was shown [61]. Other studies offer conflicting evidence with regard to activation of the UPR by AgNPs. For example, Simmard et al. showed that AgNPs increased pPERK; however, treatments that induced the highest levels of peIF2a and its downstream effectors ATF4 and CHOP did not correlate with treatments that induced the highest levels of pPERK, possibly indicating other drivers of eIF2a phosphorylation [20]. In the same report, the effects of AgNPs on pIRE were heterogeneous and were not dose-dependent. ATF6 cleavage was not examined, but in subsequent studies, they observed that AgNPs increased degradation of ATF6 [21]. We previously performed proteomic analysis in lung cancer cells with differing sensitivities to AgNPs [45]. We found that in AgNP-sensitive lung cancer cells, but not in AgNP-insensitive ones, there was a substantial decrease in overall protein synthesis following AgNP exposure. This is consistent with phosphorylation of eIF2a, which blocks translation of most mRNAs, leading to a global decrease in protein synthesis. The most affected signaling pathways in AgNP-sensitive cells were protein synthesis (including decreased activity of eIF2 and eIF4 pathways, both of which are critical for protein synthesis), protein stability, ER signaling, cell cycle, and autophagy/mitophagy. These responses are associated with mitigating proteotoxicity and, therefore, these data provide support for proteotoxic stress as a key feature of AgNP toxicity. In our current study, we could not definitively state that all three arms of the UPR were active, nor did previous studies clearly demonstrate dependence of AgNP-induced cell death on UPR signaling. Given the heterogeneity of the responses, the only consistently identified pathway is increased peIF2A and downstream effectors, including CHOP. Therefore, we conclude that the ISR, with some degree of overlap with the UPR, is a key proteotoxic stress response to AgNP exposure.

Although proteotoxic stress increased following AgNP exposure, the precise biological features that render some cells more sensitive to AgNPs than others remain to be determined. One hypothesis is that high baseline stress on the proteostasis machinery due to synthesis, folding, posttranslational modification, and secretion of ECM proteins in TNBC cells that contributes to their increased sensitivity to small molecules that induce ER stress [4,5] could also drive sensitivity to AgNPs. Alternatively, differences in the vesicle acidification rate leading to increased intracellular release of Ag^+^ could play a role [62]. Differences in AgNP uptake may be a contributing factor, but our previous studies and those of others showed that the relative sensitivity of a specific cell line to AgNPs does not necessarily correlate with the mass of AgNPs taken up by the cells, indicating that other factors dominate [17,47].

There is considerable debate regarding the potential toxicity of AgNPs used for clinical applications. Failure to separate dissolved silver cations (Ag^+^) from AgNPs before toxicity testing likely contributes to the lack of a definitive answer. Ag^+^ is highly toxic and has a distinct cytotoxic mechanism of action compared to AgNPs; specifically, AgNPs induce proteotoxicity by a mechanism that does not depend on oxidative stress, while Ag^+^ cytotoxicity is dependent upon oxidative stress [36]. Many studies that attempted to characterize the toxicity of AgNPs failed to account for Ag^+^, making it difficult to generalize their findings [47]. Estimates from various sources indicate that Ag^+^ levels in AgNP dispersions vary from 2.6 to 5.9% to greater than 70% in the majority of lab-made and commercially produced AgNPs tested [41,63]. Not all cells are equally sensitive to AgNPs or Ag^+^, and sensitivity to one does not necessarily correlate with sensitivity to the other [17,36]. Contamination of AgNP suspensions with as little as 1% mass fraction of Ag^+^ alters the toxicity profile of AgNPs [36]. Importantly, we previously showed that the AgNPs used for the experiments presented here contained less than 0.001% Ag^+^ by mass, even after 1 month of storage [36]. Another issue clouding the AgNP toxicity debate is that in vivo testing may use doses that are either sub-therapeutic (too low) or excessively high. Studies using PVP-coated AgNPs for treatment of cancer [22] or viral diseases [64] showed that doses in the range of 4–6 mg/kg delivered systemically for multiple weeks induced therapeutic responses, indicating that this range is reasonable for evaluation of the potential toxic effects of AgNPs. Direct comparison of the toxicity of citrate-coated AuNPs (10 mg/kg intravenous (IV) weekly for 8 weeks) and citrate-coated AgNPs (5 mg/kg IV weekly for 8 weeks), which roughly corresponds to a 1:1 ratio of silver and gold atoms, indicated that neither AuNPs nor AgNPs elicited any overt toxicity, though tissue discoloration and enlarged spleens were apparent [65]. Rats receiving intravenous injections of citrate-coated AgNPs (6 mg/kg for 28 consecutive days) exhibited no dose-limiting toxicities, though transient effects on liver and immune cell function were noted [66]. Metabolomic studies in mice injected with polyethylene-glycol-coated AgNPs (8 mg/kg IV) found no evidence of dose-limiting toxicity, but modest effects on liver function were observed [67]. These effects were transient and not indicative of persistent liver injury. Studies in humans receiving intravenous AgNPs are limited. However, a recent clinical trial described the use of AgNPs for prevention of severe illness and death due to COVID-19 [68]. Here, 40 patients were injected intravenously with 1.8 mg of AgNPs for 3 consecutive days (combined with standard COVID-19 treatments), and the group receiving AgNPs had significantly greater survival rates and spent fewer days on supplemental oxygen than those who did not receive AgNPs. Importantly, all of the in vivo studies described above used AgNPs with minimal contamination of Ag^+^. Taken together, these studies indicate that within the therapeutic dose range, AgNPs exhibit low off-target toxicity in vivo.

Overall, our findings provide insight into the mechanism by which AgNPs selectively kill TNBCS. Our data implied that HSPs and ISR are essential for mitigating AgNP-induced damage, pointing to accumulation of misfolded proteins as a driver of AgNP-induced cell death. At present, AgNPs, KRIBB11, and salubrinal are not used clinically, and additional studies are needed to determine their safety and utility for cancer therapy. Research defining the drivers of sensitivity to AgNPs will enable selection of cancer patients who will benefit most from AgNP exposure and will help guide future efforts to identify potential synergistic drug partners that would be beneficial for use with AgNPs for cancer therapy.

## Figures and Tables

**Figure 1 nanomaterials-14-01564-f001:**
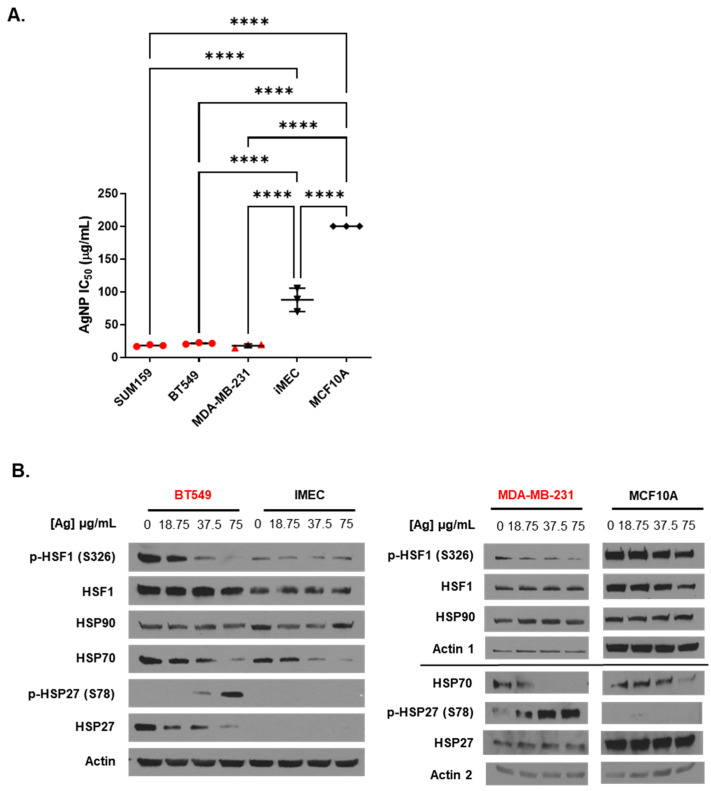
Low doses of AgNPs are more cytotoxic and modulate levels of heat shock proteins to a greater degree in TNBC cells than in immortalized mammary epithelial cells. (**A**) The cytotoxic IC_50_ values calculated from triplicate 10-point dose–response curves following 72 h exposure of each cell line to AgNPs are shown. Data points are in red for TNBC cell lines and in black for immortalized mammary epithelial cells. IC_50_ values for MCF10A cells were in excess of the dose range of AgNPs tested but were at least 200 µg/mL, as indicated in the figure. Statistical analysis was performed by one-way ANOVA followed by the post hoc Tukey test. Significant differences in IC_50_ values are indicated (**** *p* < 0.0001). (**B**) Heat shock protein expression was examined by Western blot 24 h after cells were exposed to AgNPs. Results are representative of duplicate independent experiments.

**Figure 2 nanomaterials-14-01564-f002:**
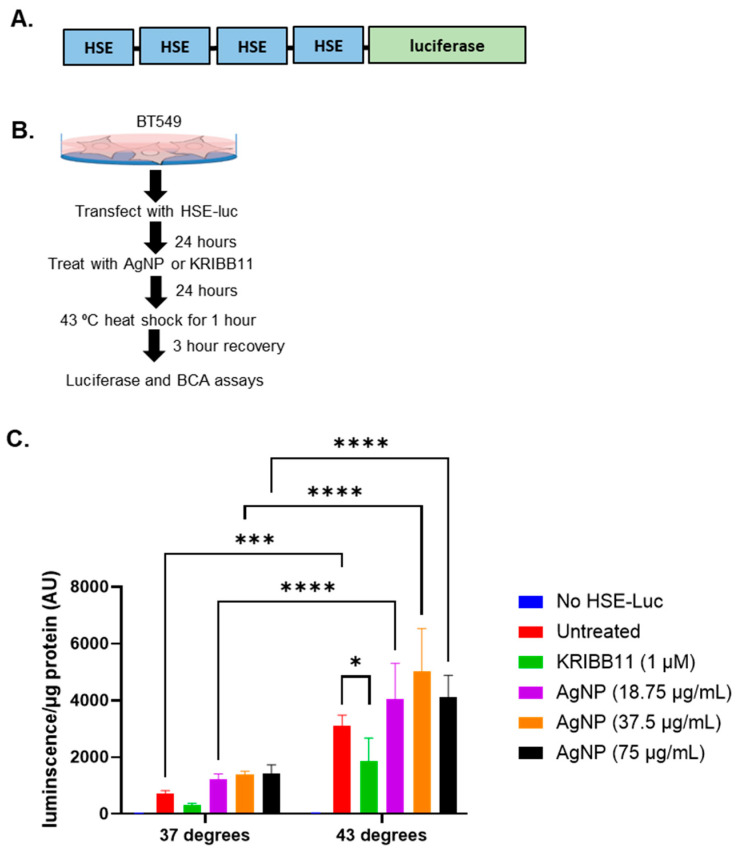
AgNP exposure does not induce a heat shock response at the transcriptional level or inhibit the heat shock response to hyperthermia. (**A**) A schematic is shown of the plasmid vector containing 4 copies of heat shock sequence elements (HSE), which drive transcription of a luciferase reporter that was used for these studies. (**B**) A schematic is shown for the experimental design of experiments to determine the impact of AgNPs on the transcriptional regulation of the heat shock response. (**C**) Luciferase expression indicative of a pHSF1-driven heat shock response is shown for cells treated as described in (**B**). Results are representative of at least four biological replicates per condition and duplicate independent experiments. Statistical analysis was performed by two-way ANOVA followed by the post hoc Tukey test. Significant differences are indicated (**** *p* < 0.0001, *** *p* < 0.001, * *p* < 0.05).

**Figure 3 nanomaterials-14-01564-f003:**
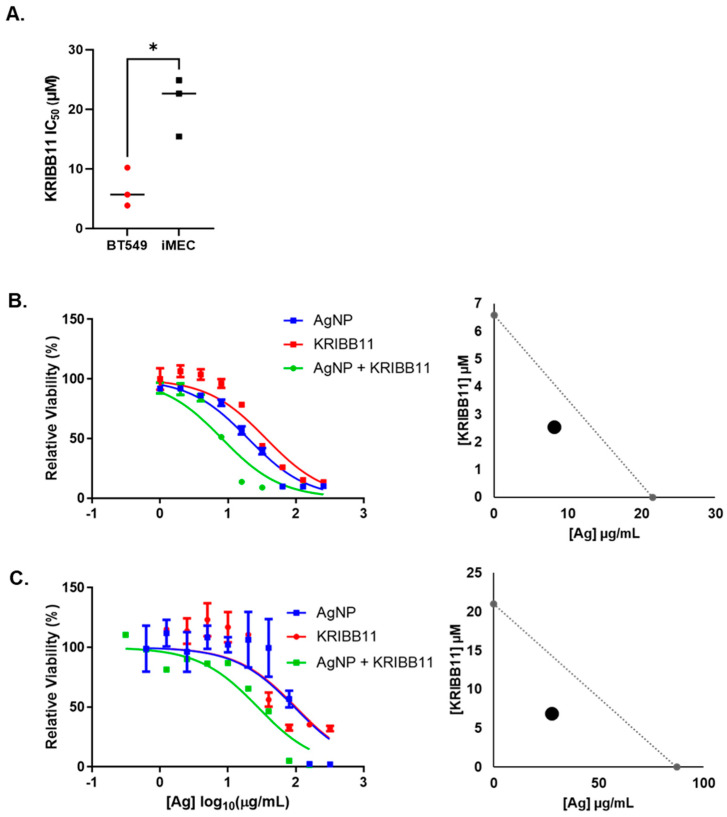
KRIBB11, an HSF1 inhibitor, synergistically increases the cytotoxicity of AgNPs in both TNBC and immortalized mammary epithelial cells. (**A**) The cytotoxic IC_50_ values calculated from triplicate 10-point dose–response curves following 72 h exposure of each cell line to KRIBB11 are shown. Statistical analysis was performed by Student’s *t*-test. A significant difference is shown (* *p* < 0.05). (**B**) BT549 or (**C**) iMEC cells were treated with AgNPs, KRIBB11, or AgNPs + KRIBB11 for 72 h and viability was assessed by the MTT assay, as shown. Isobolograms were plotted showing the expected additive effects of AgNPs and KRIBB11, and the dose of the combination that yielded Fa = 0.5 is plotted for each cell line. Results are representative of at least four biological replicates per treatment and two independent experiments per cell line.

**Figure 4 nanomaterials-14-01564-f004:**
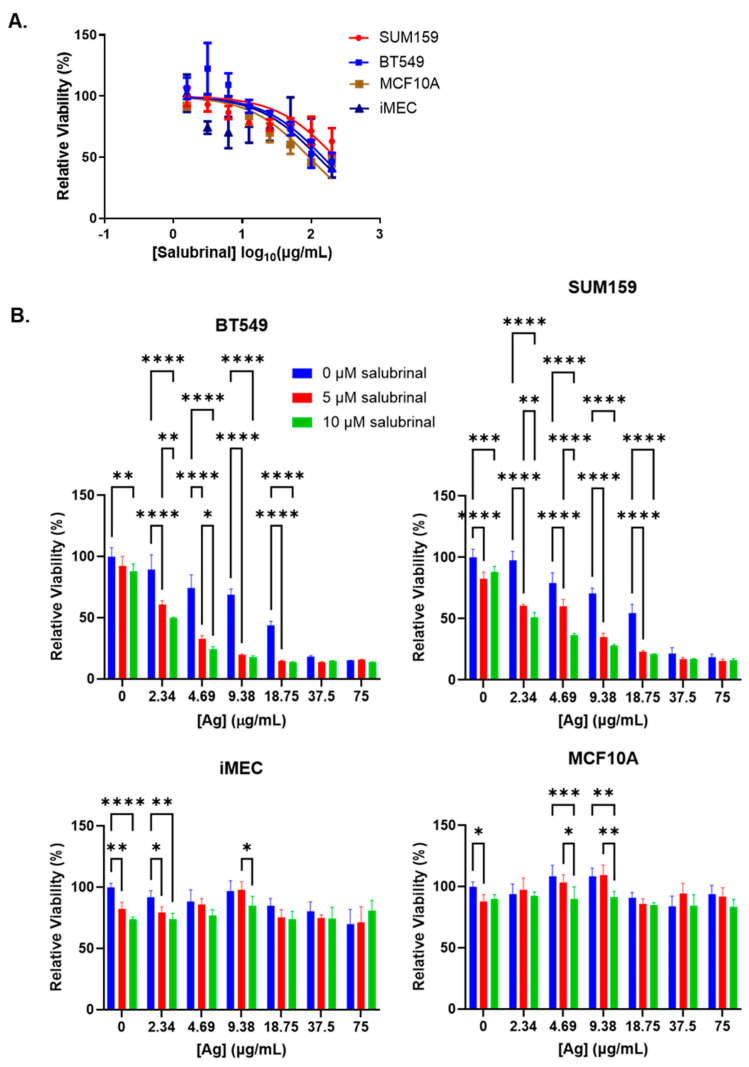
Salubrinal increases the cytotoxicity of AgNPs in TNBC but not in immortalized mammary epithelial cells. (**A**) Cells were treated with increasing doses of salubrinal for 72 h, and viability was assessed by the MTT assay. (**B**) Cells were treated with AgNPs with or without salubrinal for 72 h, and viability was assessed by the MTT assay. Results are representative of at least four biological replicates per condition and duplicate independent experiments. Statistical analysis was performed by two-way ANOVA followed by the post hoc Tukey test. Significant differences are indicated (**** *p* < 0.0001, *** *p* < 0.001, ** *p* < 0.01, and * *p* < 0.05).

**Figure 5 nanomaterials-14-01564-f005:**
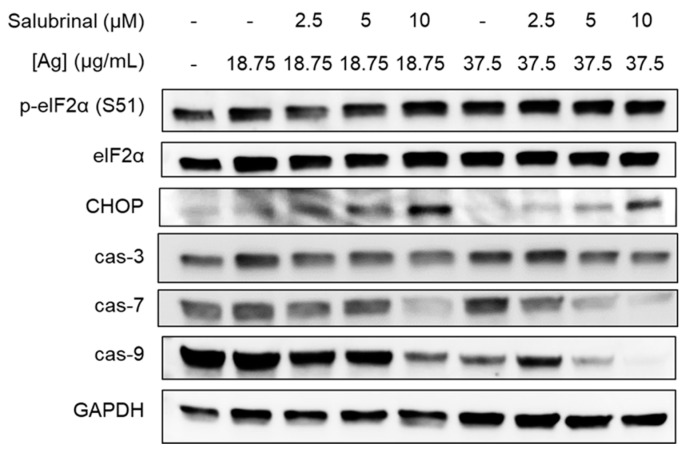
Salubrinal increases AgNP-induced ISR and apoptosis in TNBC cells. BT549 cells were treated with combinations of AgNPs and salubrinal for 24 h, and ISR and apoptosis markers were examined by Western blot. Results are representative of duplicate independent experiments.

**Figure 6 nanomaterials-14-01564-f006:**
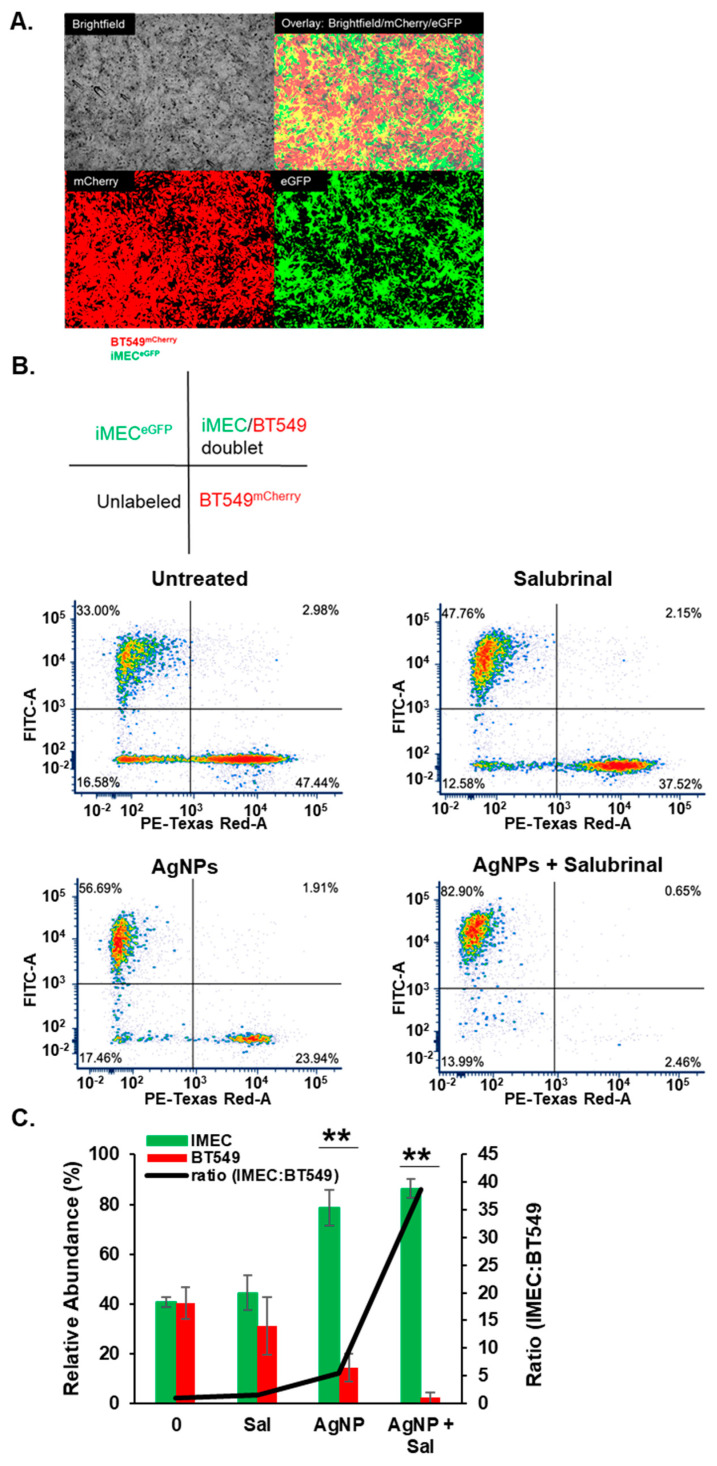
The combination of AgNPs and salubrinal selectively eliminates TNBC cells grown in co-culture with immortalized mammary epithelial cells. (**A**) BT549^mCherry^ (red) and iMEC^eGFP^ cells (green) were grown in co-culture and imaged using fluorescent microscopy. Representative images from the individual visible, red, and green light channels, and an overlay of the three channels, are shown for untreated cells. (**B**) BT549^mCherry^ and iMEC^eGFP^ cells grown in co-culture were treated with salubrinal alone (10 µM), AgNPs alone (37.5 µg/mL), or a combination of 10 µM salubrinal and 37.5 µg/mL for 8 days. Flow cytometry was used to quantify the relative fraction of each cell type remaining at the end of treatment. A schematic is shown to indicate the quadrants of the scattergrams below. The results are representative of triplicate independent experiments. (**C**) Data from triplicate experiments performed, as shown in (**B**), were quantified to show the mean percentage of iMEC^eGFP^ or BT549^mCherry^ cells (green and red bars) and the ratio of iMEC^eGFP^:BT549^mCherry^ cells (black line) detected after each treatment. Statistical analysis was performed by two-way ANOVA followed by the post hoc Tukey test. Significant differences are indicated (** *p* < 0.01).

**Table 1 nanomaterials-14-01564-t001:** Complete media formulations for cell culture.

Cell Line	Media Formulation
SUM-159	HAM’s F12 supplemented with penicillin (250 units/mL), streptomycin (250 μg/mL), 2 mM L-glutamine, 5 μg/mL insulin, 1 μg/mL hydrocortisone, 10 μM HEPES, and 5% fetal bovine serum
BT-549	RPMI supplemented with penicillin (250 units/mL), streptomycin (250 μg/mL), and 10% fetal bovine serum
MDA-MB-231	DMEM/F12 supplemented with penicillin (250 units/mL), streptomycin (250 μg/mL), 2 mmol/L L-glutamine, and 10% fetal bovine serum
MCF-10A	DMEM/F12 supplemented with penicillin (250 units/mL), streptomycin (250 μg/mL), 2 mM L-glutamine, 10 μg/mL insulin, 20 ng/mL EGF, 0.5 μg/mL hydrocortisone, 100 ng/mL cholera toxin, and 5% heat-inactivated horse serum
iMEC	DMEM/F12 supplemented with 10 µg/mL insulin, 20 ng/mL hEGF, 0.5 μg/mL hydrocortisone, and 10% fetal bovine serum

## Data Availability

Raw data is available upon request to the senior author, R.N.S.

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
