# Peer review of "Enhancement of Triple-Negative Breast Cancer-Specific Induction of Cell Death by Silver Nanoparticles by Combined Treatment with Proteotoxic Stress Response Inhibitors"

_nanomaterials, 2024, doi:10.3390/nano14191564_

Round 1

Reviewer 1 Report

Comments and Suggestions for Authors

Manuscript ID nanomaterials-3198126

Type Article

Title

TRIPLE NEGATIVE BREAST CANCER-SPECIFIC INDUCTION OF CELL DEATH BY SILVER NANOPARTICLES IS ENHANCED BY COMBINED TREATMENT WITH INTEGRATED STRESS RESPONSE INHIBITORS

The manuscript is well structured and organized,

The manuscript shows that combined treatment of integrated stress response inhibitors with silver nanoparticles selectively induces cell death in a triple-negative cancer cell model.

1) The results of the authors of the manuscript show clear results of the combined effect of inhibitors and silver nanoparticles. The authors mention that this is due to sustained proteotoxic stress in TNBC cells. However, the evidence does not show unequivocally that this is the mechanism for the differential effect.

2) Silver nanoparticles have a direct and immediate effect on the endoplasmic reticulum, and this process has direct implications for the unfolded and folded protein formation response. Could this mechanism be an alternative explanation to the authors' results.

3) Alternatively, it could be that TNBC cells have a different metabolism to immortalized epithelial cells and the processing of nanoparticles is differential, for example nanoparticles in epithelial cells are metabolized more rapidly and therefore the cytotoxic effect is not seen. Some questions to answer, what is the half-life of nanoparticles in different cell lines?

4) Previous studies had shown a differential effect of silver nanoparticles between TNBC and non-malignant breast epithelial cells (J S. et al. 2019 FASEB Bioav), and the mechanism the authors explain is by a depletion of cellular antioxidants causing extreme endoplasmic reticulum stress in TNBC type cells, but not in control cells.  It would be important to include this topic in the discussion of the manuscript in the context of heat shock proteins.

Author Response

We thank the reviewer for their critical review of our manuscript and have worked to address their comments. Below are our point by point responses to each issue the reviewer brought to our attention. Additionally, we noted several duplicate references in the manuscript and have now fixed the issue. Lastly, we performed minor text editing to improve clarity. All changes except for renumbering references have been highlighted in the revised manuscript.

Reviewer 1:

1) “The results of the authors of the manuscript show clear results of the combined effect of inhibitors and silver nanoparticles. The authors mention that this is due to sustained proteotoxic stress in TNBC cells. However, the evidence does not show unequivocally that this is the mechanism for the differential effect.”

2) “Silver nanoparticles have a direct and immediate effect on the endoplasmic reticulum, and this process has direct implications for the unfolded and folded protein formation response. Could this mechanism be an alternative explanation to the authors' results.”

Response:

These comments are inter-related and therefore we will address them together. As the reviewer notes, it is difficult to unequivically state that proteotoxicity is the only driver of AgNP induced cell death, but a growing body of literature by us and others strongly suggest that proteotoxicity contributes to AgNP induced cell death. The UPR is one form of stress response to proteotoxicity, and there is overlap between the UPR and the ISR because both can be PERK-driven. We have expanded the discussion on page 14 of the revised manuscript to provide a detailed analysis of the potential role of the UPR and a justification for our prefered usage of the ISR in relation to AgNP induced stress responses.

Reviewer 1:

3) “Alternatively, it could be that TNBC cells have a different metabolism to immortalized epithelial cells and the processing of nanoparticles is differential, for example nanoparticles in epithelial cells are metabolized more rapidly and therefore the cytotoxic effect is not seen. Some questions to answer, what is the half-life of nanoparticles in different cell lines?”

Response:

We agree that it is still unknown why some cells are more sensitive to AgNPs tan others, and differences in metabolic processes could play a role. We have added an additional paragraph to page 14 of the revised manuscript describing some posible reasons for these differences.

Reviewer 1:

4) “Previous studies had shown a differential effect of silver nanoparticles between TNBC and non-malignant breast epithelial cells (J S. et al. 2019 FASEB Bioav), and the mechanism the authors explain is by a depletion of cellular antioxidants causing extreme endoplasmic reticulum stress in TNBC type cells, but not in control cells.  It would be important to include this topic in the discussion of the manuscript in the context of heat shock proteins.”

Response:

This was one of our previous manuscripts which showed that AgNPs increased oxidized and decreased reduced forms of glutathione and NADPH as well as induced evidence of ER stress (shift in PERK mobility indicative of phosphorylation, increased GRP78, increased peIF2a/eIF2a ration, increased CHOP) in TNBC cells but not normal breast cells. However, the claim that depletion of cellular antioxidants caused endoplasmic reticulum stress is not supported by our previous data nor do we make this claim. In fact, the changes, while statistically significant, were not substantial enough for us to conclude that antioxidant depletion was directly responsible for AgNP induced cell death.

Reviewer 2 Report

Comments and Suggestions for Authors

This article presents silver nps-induced cancer cell death in combination with integrated stress response inhibitors. The topic of the presented studies is interesting and valuable. Results are well-presented and investigated. The experiments are appropriately designed and performed. Maybe I am disappointed that the authors did not synthesise nanoparticles; however, the major part of the articles is focused on biological studies, so I understand it.

I suggest improving the quality of Figure 6A ( the description size is too small; use larger and bold font). In my opinion, authors should consider changing the title. Generally, the title of the manuscript should be presented as a nominal sentence (without a finite verb).

I suggest also adding a space between the text and footnote reference.

Finally, I think this article is interesting and should be published in Nanomaterials.

Author Response

We thank the reviewer for their critical review of our manuscript and have worked to address their comments. Below are our point by point responses to each issue the reviewer brought to our attention. Additionally, we noted several duplicate references in the manuscript and have now fixed the issue. Lastly, we performed minor text editing to improve clarity. All changes except for renumbering references have been highlighted in the revised manuscript.

Reviewer 2:

1) “I suggest improving the quality of Figure 6A ( the description size is too small; use larger and bold font). In my opinion, authors should consider changing the title. Generally, the title of the manuscript should be presented as a nominal sentence (without a finite verb). I suggest also adding a space between the text and footnote reference.”

 Response:

We have made the suggested changes to Figure 6, to the manuscript title, and to the footnote formatting.